# A Universal Approach to Analyzing Transmission Electron Microscopy with ImageJ

**DOI:** 10.3390/cells10092177

**Published:** 2021-08-24

**Authors:** Jacob Lam, Prasanna Katti, Michelle Biete, Margaret Mungai, Salma AshShareef, Kit Neikirk, Edgar Garza Lopez, Zer Vue, Trace A. Christensen, Heather K. Beasley, Taylor A. Rodman, Sandra A. Murray, Jeffrey L. Salisbury, Brian Glancy, Jianqiang Shao, Renata O. Pereira, E. Dale Abel, Antentor Hinton

**Affiliations:** 1Department of Internal Medicine, Carver College of Medicine, University of Iowa, 375 Newton Rd, Iowa City, IA 52242, USA; jacob-lam@uiowa.edu (J.L.); salma-ashshareef@uiowa.edu (S.A.); renata-pereira@uiowa.edu (R.O.P.); DRCAdmin@uiowa.edu (E.D.A.); 2Fraternal Order of Eagles Diabetes Research Center, University of Iowa, 375 Newton Rd, Iowa City, IA 52242, USA; 3National Heart, Lung and Blood Institute, National Institutes of Health, 9000 Rockville Pike, Bethesda, MD 20892, USA; prasannakatti.katti@nih.gov (P.K.); brian.glancy@nih.gov (B.G.); 4Daniel K. Inouye College of Pharmacy, University of Hawaii at Hilo, 200 West Kawili St, Hilo, HI 96720, USA; mbiete@hawaii.edu (M.B.); kneikirk@hawaii.edu (K.N.); 5Department of Molecular and Cell Biology, University of California Berkeley, 142 Weill Hall, Berkeley, CA 94720, USA; margaretmungai24@gmail.com; 6Department of Molecular Physiology and Biophysics, Vanderbilt University, 2201 West End Ave, Nashville, TN 37235, USA; egarzalopez@gmail.com (E.G.L.); zervue@gmail.com (Z.V.); heather.k.beasley@vanderbilt.edu (H.K.B.); tarabia0702@gmail.com (T.A.R.); 7Microscopy and Cell Analysis Core Facility, Mayo Clinic, 200 First Street SW, Rochester, MN 55905, USA; christensen.trace@mayo.edu (T.A.C.); salisbury@mayo.edu (J.L.S.); 8Department of Cell Biology, School of Medicine, University of Pittsburgh, 3550 Terrace St., Pittsburgh, PA 15213, USA; smurray@pitt.edu; 9Department of Biochemistry and Molecular Biology, Mayo Clinic, 200 First Street SW, Rochester, MN 55905, USA; 10National Institute of Arthritis and Musculoskeletal and Skin Diseases, National Institutes of Health, 9000 Rockville Pike, Bethesda, MD 20892, USA; 11Central Microscopy Research Facility, University of Iowa, Iowa City, IA 52242, USA; jian-shao@uiowa.edu

**Keywords:** cristae, image analysis, Mitochondria Endoplasmic Reticulum Contacts (MERCs), image processing, ImageJ, TEM analysis, TEM quantification, mitochondria

## Abstract

Transmission electron microscopy (TEM) is widely used as an imaging modality to provide high-resolution details of subcellular components within cells and tissues. Mitochondria and endoplasmic reticulum (ER) are organelles of particular interest to those investigating metabolic disorders. A straightforward method for quantifying and characterizing particular aspects of these organelles would be a useful tool. In this protocol, we outline how to accurately assess the morphology of these important subcellular structures using open source software *ImageJ*, originally developed by the National Institutes of Health (NIH). Specifically, we detail how to obtain mitochondrial length, width, area, and circularity, in addition to assessing cristae morphology and measuring mito/endoplasmic reticulum (ER) interactions. These procedures provide useful tools for quantifying and characterizing key features of sub-cellular morphology, leading to accurate and reproducible measurements and visualizations of mitochondria and ER.

## 1. Introduction

Organelles involved in cellular metabolism are some of the most studied subcellular structures. Mitochondria play essential roles in multiple metabolic pathways and are responsible for generating cellular energy. Mitochondria, and the inner folds known as cristae, are involved in apoptosis and cell death pathways, as well as the production and consumption of reactive oxygen species (ROS) [1]. The endoplasmic reticulum (ER) plays a key role in protein and lipid synthesis and in calcium homeostasis [2]. The methods detailed here may also be used in the analysis of other organelles, such as Golgi apparatus, lysosomes, and autophagosomes, which contribute to the pathogenesis of many diseases, including type II diabetes mellitus, cardiovascular disease, and Alzheimer’s disease [3,4,5,6]. Due to the central role of ER, mitochondria, and cristae in the pathophysiology of a variety of disorders, studying the structure of these organelles and accurately assessing how they are disrupted under disease conditions is of paramount importance.

Transmission electron microscopy (TEM) is a widely used tool for studying organelle ultrastructure. TEM produces high-resolution images by transmitting electrons through an ultrathin section of the sample and magnifying the resulting image using a series of lenses [7]. TEM has enabled researchers to study the ultrastructure of organelles in a wide variety of healthy and diseased cells and tissues, yielding important insights into cellular processes and disease development [8].

While overall organelle structures can be readily determined from TEM images, quantifying visible changes in organelle morphology presents a substantial challenge. To address this issue, we propose a standardized approach for analyzing TEM images using the image analysis tool *ImageJ*, also known as *Fiji*, which was developed by the National Institutes of Health (NIH). In this protocol, we outline methods for measuring features of important organelles, including mitochondria and ER. In addition, we provide strategies for assessing how these organelles interact with one another. By using a straightforward method to quantify organelle morphology as proposed in this study, investigators can generate accurate and reproducible measurements of organelle features, increasing the precision and relevance of their data and the capacity to compare findings with those of other investigators.

## 2. Methods

### 2.1. Animal Care & Maintenance

Care of mice was performed based on prior protocols [9] and in accordance with protocols approved by the University of Iowa Animal Care and Use Committee (IACUC). All experiments were performed on male mice, C57Bl/6J, the majority being wildtype. Recombination was induced in 4-week-old mice utilizing intraperitoneal injection of 20 mg/kg of tamoxifen (Sigma, St. Louis, MO, USA, T5648) across 5 days to generate mOPA1 KO mice. Following this, for 4 weeks, the mice were fed standard chow (2029X Harlan Teklad, Indianapolis, IN, USA) to recover from the injections. Animals were housed at 22 °C with a 12-h light, 12-h dark cycle with free access to water and standard chow.

### 2.2. Isolation of Satellite Cells

Satellite cell isolation and differentiation were performed as described previously with minor modifications [10]. Briefly, the skeletal muscles of gastrocnemius and quadriceps were excised from C57BL/6J wildtype at 8–10 weeks of age. The muscles were washed twice with 1× PBS supplemented with 1% penicillin-streptomycin and fungizone (300 μL/100 mL). DMEM-F12 media with collagenase II (2 mg/mL), 1% penicillin-streptomycin, and fungizone (300 uL/100 mL) was added to the muscles, and the muscles were shaken for 90 min at 37 °C. The tissue washing process was repeated, using DMEM-F12 media, same specifications as before except collagenase II was changed to 0.5 mg/mL, in a shaker for 30 min at 37 °C. Following this, the tissue was cut and passed through a cell strainer (70 μm). Following centrifugation, satellite cells were plated on BD Matrigel-coated dishes. Cells were differentiated into myoblasts through the usage of a mixture of DMEM-F12, 20% fetal bovine serum (FBS), 40 ng/mL basic fibroblast growth factor (bfgf, R and D Systems, 233-FB/CF), 1× non-essential amino acids, 0.14 mM β-mercaptoethanol, 1× penicillin/streptomycin, and Fungizone. Myoblasts were maintained with 10 ng/mL bfgf and then differentiated in DMEM-F12, 2% FBS, 1× insulin–transferrin–selenium, when 80% confluency was reached.

### 2.3. Cell Culture & Differentiation

Satellite cell isolation was performed as previously described, and cell culture was performed similarly to past protocols [9,10]. Satellite cells were plated on BD Matrigel-coated dishes and activated to differentiate into myoblasts in DMEM-F12, 20% fetal bovine serum (FBS), 40 ng/mL basic fibroblast growth factor (R&D Systems, 233-FB/CF), 1× non-essential amino acids, 0.14 mM β-mercaptoethanol, 1× penicillin/streptomycin, and Fungizone. Myoblasts were maintained with 10 ng/mL basic fibroblast growth factor before they were differentiated in DMEM-F12, 2% FBS, 1× insulin–transferrin–selenium, when 85% confluency was reached. They were cultured at 37 °C, 5% CO_2_ Dulbecco’s modified Eagle’s medium (DMEM; GIBCO) supplemented with 10% fetal bovine serum (FBS; Atlanta Bio selected), and antibiotics-1% penicillin-streptomycin (Gibco, Waltham, MA, USA). Three days after differentiation, myotubes were infected with adenovirus for ntGFP or GFP-Cre for OPA1 deletion at a multiplicity of infection sufficient to infect >95% of the cells with minimal cell death. Adenoviruses were obtained from the University of Iowa Viral Vector Core facility. Experiments were performed between 3 and 7 days after infection for a total of 6 days of differentiation.

### 2.4. Thapsigargin Treatment

Fibroblasts and satellite cells were treated with Thapsigargin in accordance with prior methods [11]. Mouse fibroblasts and satellite cells were treated with Thapsigargin (2 μg mL^−1^; Sigma) for 10 h, followed by crosslinking with 1% formaldehyde for 10 min. Following this, cells were ready for imaging via the protocol above.

### 2.5. Systematic ImageJ Analysis for TEM Measurements of Organelle Morphology

To perform unbiased morphometric analysis of organelles, a team of individuals must be assembled. Ideally, one individual conducts the experiment and fixation of the murine and human cells. A second individual processes and collects pictures in a blinded and randomized manner with the electron microscope, and the last individual quantifies the anonymized samples. Their collective findings are averaged to decrease individual subjective bias. Additionally, the individuals tasked with quantification should receive randomized images of the whole cells and higher magnifications. 

#### 2.5.1. ImageJ Quantification


Quantification analysis methods were developed using the National Institutes of Health (NIH) *ImageJ* software. *ImageJ* is an open-source image processing software designed to analyze multidimensional scientific images, such as TEM and confocal microscopy data sets. Notably, the NIH *ImageJ* software utilizes pixel count to display a displayed image in 2048 by the 2048-pixel frame. Each pixel in the frame is assigned a horizontal and vertical coordinate. Any straight line on the image can be defined by two pixels on each end, pixel 1 and pixel 2. Therefore, coordinates of pixels are annotated *p*1 = (*x*1, *y*1) and *p*2 = (*x*2, *y*2) for pixels 1 and 2. Measurements needed to decipher the changes in organelle morphology were constructed using the following calculations [12,13,14,15,16,17,18,19,20,21]:**(1)** **Length (*L*):** The length is measured by applying the Pythagorean Theorem to the coordinate pixels. Thus, where *Lp*1,*p*2 denotes the Euclidean distance between pixel 1 and pixel 2. Length is given by (Equation: *L(p_1_ + p_2_) = √(x_1_ − x_2_)^2^ + (y_1_ − y_2_)^2^*)**(2)** **Area:** This is calculated similarly. The area calculated by ImageJ is the area of pixels.
**i.** **Rectangles (*^A^**rectangle*):** Let *L*1 be the length (as defined previously) of one side of the rectangle and let *L*2 be the length of the other side; the area is then given by (Equation: ^A^*rectangle* = L_1_ × L_2_)**ii.** **Circles (*A*
*circle*):** Once the circle is traced, the software chooses two opposite pixels on the circle, pixel 1 and pixel 2. Equation then gives the diameter *LD*. Circle area is given by (Equation: A*_circle_* = *π(L_D_*/2)^2^)**(3)** **Perimeter:****i.** **Rectangles (*P*
*rectangle*):** Consider *L*1 and *L*2 as defined in Section 2, rectangle perimeter is given by (Equation P*_rectangle_* = 4*L*_1_ × *L*_2_)**ii.** **Circles (*P*
*circle*):** Consider LD as defined in 2.ii., circle perimeter is given by (Equation: *P_circle =_ π × L_D_*)**(4)** **Circularity Index (*C**_i_*):** Consider *P circle* and *A circle* as in 3.ii and 2.ii, respectively. The circularity index is given by (Equation: Ci = 4π × Acircle/(Pcircle)^2^)**(5)** **Volume:** Surface area of substructure features are divided by the area of the structure (SA:A) to provide an estimate volume.

#### 2.5.2. ImageJ Parameters

First, a low magnification image (in a DM3 or TIFF format) of the whole cell is uploaded to *ImageJ to quantify these many diverse measurements*. The entire cell is then divided into quadrants, facilitated by the *ImageJ* plugin called quadrant picking (https://imagej.nih.gov/ij/plugins/quadrant-picking/index.html, accessed on 17 June 2021) [22]. After splitting the first image into four quadrants, two quadrants must be randomly selected and used for each group’s complete analysis. To obtain accurate and reproducible values, these measurements should be repeated in a minimum of 10 cells with three analyses, each from a different individual. If there is variability in the three individuals’ data, we have found expanding the number of cells to 30 cells per individual effectively reduces variability. During analysis, measurements can be tracked with the ROI Manager interface (Analyze > Tool > ROI Manager. Subsequently, the necessary measures can be set (Analyze > Set Measurements: Area, Mean gray value, Min & Max gray value, Shape descriptors, integrated density, Perimeter, Fit ellipse, Feret’s Diameter).

## 3. Protocol for ImageJ Installation 

### 3.1. Installing and Preparing ImageJ Software for Analysis


i.Download *ImageJ* software from the NIH website (Download (nih.gov, accessed on 17 June 2021).ii.Install and open the ImageJ softwareiii.Open the Region of Interest (ROI) Manager, which is used to record and keep track of measurements, by selecting **Analyze**
**➧ Tools**
**➧ ROI Manager** (Figure 1A).iv.Click on **Analyze** ➧ **Set Measurements** to input the measurements that *ImageJ* will make, such as area, circularity, and perimeter (Figure 1A).v.Drag the image to be analyzed (TIFF or DM3 file) directly into *ImageJ* (Figure 1B).


Alternatively, click **File** ➧ **Open** to open the selected image.


**Considerations:**


For accuracy and reproducibility, ensure that each image contains a scale bar, bar length, and image magnification. The scale bar and bar length are important for setting the appropriate units within the *ImageJ* settings. Quantification of samples should be performed by three individuals in a randomized and blinded manner to ensure an unbiased approach.

### 3.2. Analyzing Mitochondrial Morphology


i.Measuring mitochondrial area, circularity, and perimeter.ii.Click on **Freehand Selections** on the toolbar to access the **Freehand** tool.iii.Trace the outline of the entire cell. To store the measurement, click **Add** on the ROI Manager. This will be used to normalize later measurements.iv.Trace the outer mitochondrial membrane of each mitochondrion. Add the shape to the ROI Manager (Figure 2A).v.Click on **Measure** in the ROI Manager to obtain the measurements for each shape.


### 3.3. Measuring Mitochondrial Length and Width


i.Click on Straight, Segmented, or Freehand Lines on the toolbar and select Straight Line.ii.Draw a straight line along the major and minor axis of each mitochondrion (Figure 1B).iii.Add the measurements to the ROI Manager (Figure 1B).iv.Click the **Measure** function to obtain lengths and widths.


### 3.4. Analyzing Cristae Morphology


i.Split each image into four quadrants and randomly pick two quadrants to analyze; use the same quadrants for all images.ii.Using the **Freehand** tool in *ImageJ*, outline the outer membrane of a mitochondrion.iii.Add the measurement to the ROI Manager.iv.Trace the outline of each crista within the mitochondrion. Add each measurement to the ROI Manager (Figure 2B).v.Click on **Measure** in the ROI Manager to obtain the area of each crista, and cristae surface area is the sum of the areas of all the cristae in a single mitochondrion.vi.To calculate cristae estimated volume density, cristae surface area is divided by the area of the mitochondrion.vii.To determine the cristae number, count the number of cristae in each mitochondrion.Alternatively, you can use the number of cristae measurements obtained earlier.


To assign a mitochondrion a cristae score, evaluate cristae abundance and form. Assign a score between 0 and 4 based on the quantitative number of cristae and their appearance (0—no sharply defined crista, 1—greater than 50% of the mitochondrial area without cristae, 2—greater than 25% of mitochondrial area without cristae, 3—many cristae (over 75% of area) but irregular, 4—many regular cristae) [23].

Considerations: 

Average cristae measurements in cells are based on 1000 total mitochondria from 100 cells (to average 10 mitochondria per cell) or 10 mitochondria per cell from 2000 cells (to total 20,000 mitochondria) [12]. For tissue samples, average measurements are based on three independent animals with 330 mitochondria measured from at least 10 images.

### 3.5. Analyzing Mitochondria-ER Contacts (MERCs)


i.Split each image into four quadrants and randomly pick two quadrants to analyze; use the same quadrants for all images.ii.Identify a Mitochondrion-ER contact (MERC), a mitochondrion in close contact with the ER membrane, in your TEM image.iii.Using the **Freehand Selections** tool, trace both the outer membrane of the mitochondrion and the ER membrane at a contact point (Figure 2C).iv.To obtain the MERC contact length, click on the **Straight, Segmented, or Freehand Lines** tool on the toolbar, select **Freehand Line**, and draw a line spanning the length of the contact site (Figure 2C).v.Add the measurement to the ROI Manager and use the **Measure** function to determine the length of the contact.vi.To measure the MERC distance, use the **Freehand Line** tool to draw a line between the two organelles (Figure 2D).vii.Add the measurement to the ROI Manager and use the **Measure** function to determine the distance.viii.To calculate percent coverage, divide the contact length by either the mitochondrial surface area (mitochondrial percent coverage) or ER surface (ER percent coverage), and multiply the value by 100 to obtain a percentage.Similar measurements can be made for other organelle interactions using the same steps detailed here.


Considerations:

Average MERC changes in cells should be based on 100 to 400 images. For tissue samples, measurements should be based on three independent animals with 100 to 400 images per animal. Average MERC distance and length in cells should be based on 50 to 100 images. For tissue samples, measurements should be based on three independent animals with 10 to 15 images per animal.

### 3.6. (Optional) Pseudo-Coloration of Organelles Using ImageJ


i.Open ImageJii.Select **File**
**➧ Open** and select a desired TIF (or alternative) image.iii.Select **Image**
**➧ Type**
**➧ RGB Color** to allow for visible colors (Appendix A).iv.Select the **Color Picker Tool** from the toolbar and select the desired color for the organelle (Appendix A)v.Outline the organellevi.On the right-hand side of the tool bar, select the **Paintbrush Overlay Tool**.vii.Double click the **Paintbrush Overlay Tool** and set the width to 1, or a small enough width to outline, and the transparency to 0 (Appendix A)viii.Trace the outlines of the organelles with the brush tool and mouse (Appendix A).ix.Use the +/− buttons to better zoom in to the picture as needed.


If mistakes are made, **edit**
**➧ Undo** can undo only the last action.
x.Color the organellexi.Double click the Paintbrush Overlay Tool and set the width to 3–15, depending on the size of the organelle to color in.xii.Use the mouse to fill in the previously made outline (Appendix A).


Completing the image
xiii.Repeat Steps 5.1.5 and 5.1.6 with new colors as many times as necessary for each organelle or structure that needs to be colored (Appendix A).



Saving the image


To save the overall image, click **File**
**➧ Save.**

To save only the colored areas, click **Image**
**➧ Overlay**
**➧ To ROI Manager,** and from the ROI manager, these colored areas can be saved through selecting all the applicable colored areas and selecting **More**
**➧ Save** (Appendix A)**.**

Considerations:

When doing overlays, *ImageJ* only allows for one action to be undone, so working too quickly may result in an incorrect structure being saved. Although there are tools for automatically filling in the outline in *ImageJ*, these tools were not consistently reliable and/or accurate in our studies.

When dealing with overlays in *ImageJ*, the *ImageJ* magnification changes the scale of the coloring; therefore, once an ideal zoom amount has been selected, it should not be altered.

### 3.7. (Optional) Pseudo-Coloration of Organelles Using Adobe Photoshop


i.Install and Open Adobe Photoshopii.Click **File** ➧ **Open** and select the saved *ImageJ* export file.iii.Under the **Layers** menu click **New** ➧ **Layer** and ensure the new layer is selected and above the layer with the image.iv.Outline the organelle (Appendix A).v.On the left tool sidebar, select the **Freeform Pen Tool.** Use tiny increments to slowly outline. If mistakes are made, use the eraser tool to fix them.Ensure one continuous line is used for the outline of organelles.vi.Right-click the line drawn and click **Stroke Path**, set to pencil or paintbrush to create a continuous and clean line (Appendix A).vii.Right-click the original line drawn and click **Delete Path** to remove the original line and only be left with the clean version (Appendix A).viii.Clean up the outlineix.On the left tool sidebar, select the **Eraser**.x.Use the eraser to manually erase parts where the lines are uneven or unrepresentative of the organelle.xi.Under the **Layers** menu, click **New** ➧ **Layer** and ensure the new layer is selected and below the layer with the outline.xii.Color the organellexiii.On the left-hand side, select the **Paint Brush** tool, and at the bottom select a color Appendix A).xiv.Use the mouse to color in the area outlined. This does not need to be precise as it will later be adjusted (Appendix A).xv.To fix uneven coloring, once fully colored, for each organelle, select the **Magic Wand**. Go to the outline layer and click onto each organelle to cause the coloration to fit evenly inside the outline (Appendix A).xvi.Delete the Color layer, so the only layer left is the Outline layer with the new colors and the background image (Appendix A).


Considerations:

During outlining the organelle, one can do small lines and create a stroke path for each line, this increases accuracy but will take a longer time.

It is recommended that one saves continuously during this process as it is easy to begin working on the wrong layer.


xvii.During the coloration process, so long as the outline is continuous, one can use the **Magic Wand** from the tool menu to instantly fill in the area with one color.xviii.Click **File** ➧ **Save As…** and save as a desired export file for further analysis, if necessary.


## 4. Representative Results

This protocol standardizes an approach to obtaining accurate and reproducible measurements of organelle morphological features. Below, we discuss findings based on this TEM image analysis approach.

### 4.1. OPA1 Knockdown Decreases Mitochondrial Coverage

Optic atrophy protein-1 (OPA1) is an inner mitochondrial membrane protein that plays an essential role in inner mitochondrial membrane fusion, in concert with mitofusin proteins MFN1 and MFN2 that mediate outer membrane fusion [24]. Previous studies have demonstrated that OPA1 knockdown inhibits mitochondrial fusion and increases the number of fragmented mitochondria [12]. Using our proposed method outlined above, we confirmed this finding in mouse skeletal muscle tissue (Figure 3A). Specifically, we showed that knocking down OPA1 significantly decreases the mean mitochondrial area for the perinuclear mitochondria cluster (Figure 3B). Additionally, we observed an increase in both the number of mitochondria per square micron and the average circularity index (Figure 3C,D). These findings suggest that loss of OPA1 decreases mitochondrial fusion and increases fission events, consistent with previous studies.

### 4.2. OPA1 Effects on Cristae Structure

Beyond its role in mitochondrial dynamics, OPA1 plays a direct role in cristae remodeling [25]. To assess the effect of OPA1 on cristae morphology, we used our TEM analysis protocol to quantify cristae changes in OPA1-knockdown skeletal muscle tissue (Figure 4A). Disruption of cristae morphology was confirmed, and reductions were observed in the average number of cristae per mitochondrion, average cristae surface area, and cristae volume density in OPA1-knockdown cells (Figure 4B–D). Furthermore, the cristae score was reduced in OPA1-knockdown cells relative to the control group (Figure 4E). These results confirm that OPA1 is essential for cristae remodeling and that loss of OPA1 disrupts normal cristae structure. Loss of such cristae structure may be explored in respect to the pathogenesis of metabolic diseases.

### 4.3. Thapsigargin Treatment Increases MERCs

The interaction points between the ER and mitochondria, commonly referred to as mitochondria-ER contacts (MERCs), are becoming more of a focus of study in cellular biology. Emerging research has implicated defective MERCs in multiple age-associated diseases, such as neurodegenerative diseases, metabolic syndrome, and cardiovascular diseases [26]. To test our method of quantifying MERCs, we measured the effect of thapsigargin treatment or vehicle/control, dimethyl sulfoxide (DMSO) on MERCs in murine myotubes (Figure 5A). Thapsigargin is a sarco-ER Ca^2+^- ATPase (SERCA) inhibitor that induces ER stress [27]. ER stress inducers like thapsigargin causes a narrowing of the space between mito and ER [27]. In our analysis, we found that thapsigargin treatment greatly decreases the distance between mitochondria and the ER compared with the control cells (Figure 5B). The average contact length is also significantly increased in treated cells (Figure 5C). These findings suggest that thapsigargin treatment influences MERC structure by increasing the degree of contact between mitochondria and the ER and decreasing the distance between the two organelles, as previously reported [28].

## 5. Discussion

TEM is one of the most powerful tools for assessing organelle morphology in cells and tissue. Currently, there is a method to analyze mitochondria or organelle morphology that can be done by point counting [29]. Point counting functions by dividing the image into a standardized grid, and calculating valuable metrics such as organelle volume and area. One advantage of using point counting is that it is systematically done, so it is more accurate. However, the proposed protocol allows for the quantification of finer structural details, including mito-ER interactions and cristae measurements, which has been a limitation to point counting. Taken together, using this protocol allows for the advancement of cristate measurements.

Herein, we describe a straightforward approach for accurately quantifying organelle features from TEM images, which can easily be reproduced in all cell and tissue types. Using this method, we verified results from multiple previous studies that assessed how manipulation of mitochondrial and ER proteins using genetic and pharmacological approaches alters organelle morphology. On a broader scale, this method can be applied to any field of study and cell type where organelle structure is a focus. For example, mitochondria play a role in many complex diseases, including type II diabetes, cardiomyopathy, and Alzheimer’s disease [4,6]. Therefore, understanding how these diseases impact mitochondrial structure through the use of TEM in conjunction with this precise methodology may lead to a greater understanding of their pathophysiology.

One critical step for successfully implementing this method is to ensure that the image to be analyzed is clear enough to identify specific structures. Several organelles, including mitochondria, have distinguishing characteristics and are easily observable even at lower magnifications. However, finer structures, such as the cristae within mitochondria, may not be so readily apparent. To address this, higher magnifications may be used to better resolve organelles of interest. This protocol additionally details the methods of pseudo-coloring through either Adobe Photoshop or *ImageJ*. Pseudo-coloration is typically used to look at spatial relationships and to distinguish individual features by color, however, it is also a useful tool to accentuate finer subcellular details, such as cristae within mitochondria. Although TEM is a useful and powerful tool for assessing organelle structure in cells and tissues, this method of structural analysis has some limitations. Individual proficiency with using the tools of *ImageJ*, especially the freehand tools, may lead to variability in the data, or the amount of time required to complete the analysis. However, most investigators should be able to take advantage of this method with practice. Another limitation is that samples must be sliced into ultrathin segments to be properly imaged, meaning three-dimensional characteristics cannot be fully captured by this technique. Thus, features such as organelle volume or organelle interactions in three dimensions cannot be reliably assessed without using alternative methods. Point counting is a useful and accurate method to also measure organelle features [29,30]. Point counting provides an estimation of organelle features [29,30]. The point method is a procedure that allows for the accurate estimation of metrics, including area and volume, by the alignment of a point grid over the microscopic images [29,30]. Point counting is an efficient manual technique that allows for one to measure in an unbiased way the estimated volume fraction of a organelle or subcellular object being studied [29,30]. Since point counting does not necessitate the quantification of each individual structure being studied, it can be done relatively more quickly than other comparable quantification methods [29,30]. As with many quantification methods, point counting is dependent on precision and accuracy of both the sample, the sample preparation, and the relative magnification [29,30]. For more details regarding point counting, please refer to Howard & Reed (2004) [29]. More advanced techniques, including focused ion beam scanning electron microscopy (FIB-SEM) or serial block-face scanning electron microscopy (SBF-SEM), can be used to image subcellular structures more accurately in three dimensions, although precise quantification is extremely time-consuming and requires computing skills [31]. A third limitation of TEM and our analysis method is the lack of live imaging. Due to the requirement of fixing samples in resin, TEM captures only a single snapshot of a sample in time. Consequently, information about dynamic characteristics, such as how organelles move in the cell and change over time cannot be obtained through this approach. Therefore, other techniques, such as fluorescent staining or proximity ligation assay, might be needed to complement this approach [32]. Despite these limitations, our TEM analysis method represents a powerful tool for accurately assessing and quantifying organelle morphology, with the potential for broad application in the study of metabolic disorders and other diseases associated with organelle disruption.

## 6. Perspectives on TEM Measurements

For organelles, the orientation will always matter. Depending on the orientation at which the measurement is taken, organelles, especially mitochondria, may have different appearances, potentially lying either along the x-axis or y-axis, complicating measurement. In this sense, it is difficult to consistently measure length or width without mixing them up, depending on whether longitudinal or lateral tissue sections were measured. Beyond this, the orientation of the mitochondria will further differ based on the tissue type. Thus, we recommend that proper orientation be taken into account or the usage of more general measurements, including area, which is not reliant on length or width measurements. Furthermore, length and width for the mitochondria may also be measured as major-axis length and minor-axis length if one is unfamiliar with the exact orientation of the mitochondria being measured. This will minimize confusion regarding whether length or width are being measured. Along with this, measurements not dependent on orientation, including Feret’s diameter and area (discussed in the methods), should also be utilized. 3D imaging techniques, including SBF-SEM, may be utilized to analyze the length and width of mitochondria more accurately. 

## 7. Perspectives on TEM Standardization

Uniform randomization is critical for the unbiased quantification of TEM images [29,33,34,35]. However, even when prioritizing randomization, bias can still happen. For example, a researcher may unconsciously choose images that would showcase more pertinent or presentable data. Unless every single section of the tissue or cell has a perfectly equal chance of being imaged, such as through stereology, some amount of bias in measurement may occur [29,33,34,35]. To avoid both conscious and unconscious bias, the optimal option would be to either employ an algorithm to randomly select tissue sections or employ time-consuming alternative methods to image the entire sample, such as volume reconstruction [35]. However, bias could still occur during the sampling stages. Biased data is not easily recognizable as it may either be accurate or inaccurate and precise or imprecise, meaning there is no uniform way to recognize data that has been skewed [29]. As such, it is vital for randomness and uniformity to be adopted at every step of the procedure, from animal processing to data analysis, to ensure that any biases in the results are minimized. 

We did this in several ways; to begin with, once mice finished growing, mice were assigned new identification numbers to prevent processing bias, and processing was done by a different individual. Then, mice were euthanized and harvested for gastrocnemius muscle following the randomized procedure previously laid out [35]. The muscle was cut into five even rows with small surgical scissors before being cut into 1 cm^3^ sections by a razor blade From each row, three random pieces were placed in glutaraldehyde fixative and then transferred to the next blinded person in the laboratory. This person was responsible for making five grids from each sample and taking images in a randomized way by providing low-magnification, mid-magnification, and high-magnification images of multiple areas, with additional pictures for each fiber. After pictures were taken, they were given randomized sample numbers. The images were then loaded onto a server where three independent individuals analyzed the data sets using the protocol we laid out. Data was finally given back to the original person and the PI, at which point it was finally unmasked and plotted. Through utilizing these methods to minimize bias, each person that does part of the sampling and measuring of the TEM images is separate and completely blind to sample conditions.

Previous sources have discussed how random sampling must occur, as if sampling is not random, neither will the statistical analysis be, even if verified methods such as stereology and point counting are performed [29,35]. To ensure randomness, at every point of the protocol, the samples were randomized and given to a blinded individual. Furthermore, we ensure that all the analyses are being done by someone simply following the methodology. We utilize the well-documented method of stereology during image sectioning to ensure that we minimize bias [29,35]. From there, analyses are performed by three separate individuals to ensure accurate and free-bias measurement. Additionally, throughout the measurement process, we ensure a high sample number (n) by measuring a total of 1000 mitochondria either by examining 10 mitochondria per cell from 100 cells or 100 mitochondria per cell from 10 cells. Finally, internal controls are performed with these quantifications and known references, including insulin treatment in muscle cells [12] or OPA1 knockout muscle cells [33,34], to ensure that results are reliably being reproduced, even when analyzed with different methods. These systematic changes allow for overcoming bias by being careful and ensuring randomization throughout the whole process, while also avoiding shortcomings of point counting regarding ultrastructure measurement.

## Figures and Tables

**Figure 1 cells-10-02177-f001:**
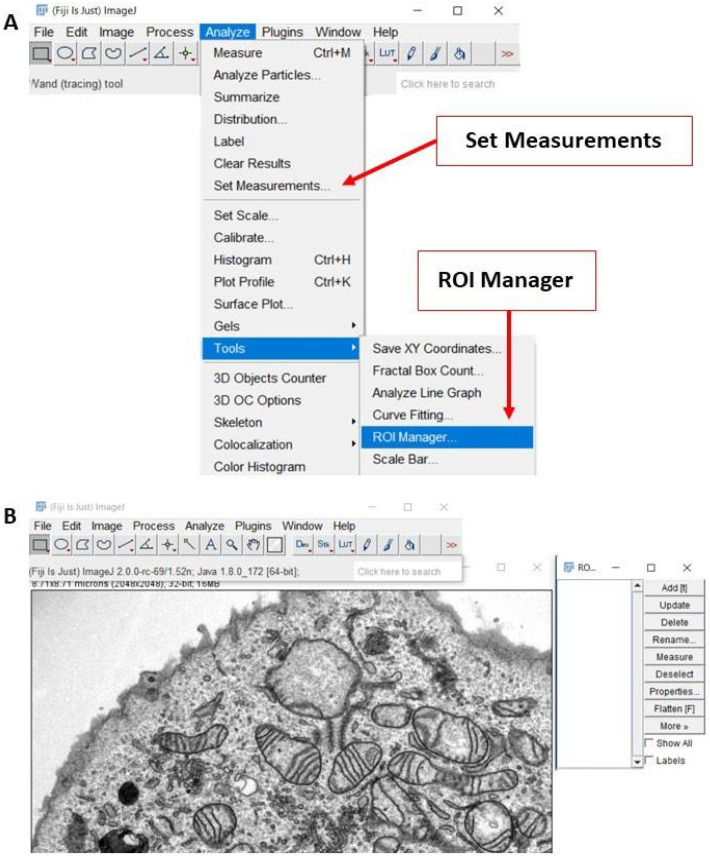
**Preparing *ImageJ* software for image analysis.** (**A**) On the *ImageJ* toolbar, the “Analyze” menu contains many of the settings and tools needed for this analysis method. (**B**) Representative screenshot of a transmission electron microscopy (TEM) image that is ready to be analyzed. The ROI menu is shown to the right TEM image.

**Figure 2 cells-10-02177-f002:**
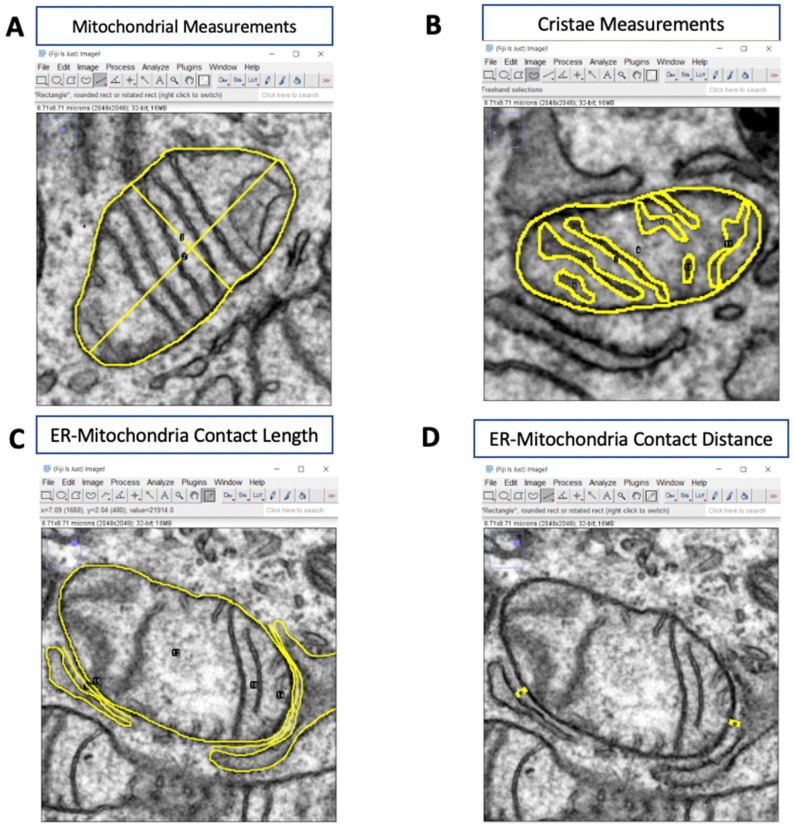
**Analyzing organelle morphology with *ImageJ*.** (**A**) Representative TEM image illustrating how to measure mitochondrial length, width, area, and circularity. (**B**) Representative TEM image illustrating how to obtain cristae measurements (yellow). (**C**,**D**) Representative TEM images illustrating how to determine ER-mitochondria contact length and ER-mitochondria contact distance (yellow).

**Figure 3 cells-10-02177-f003:**
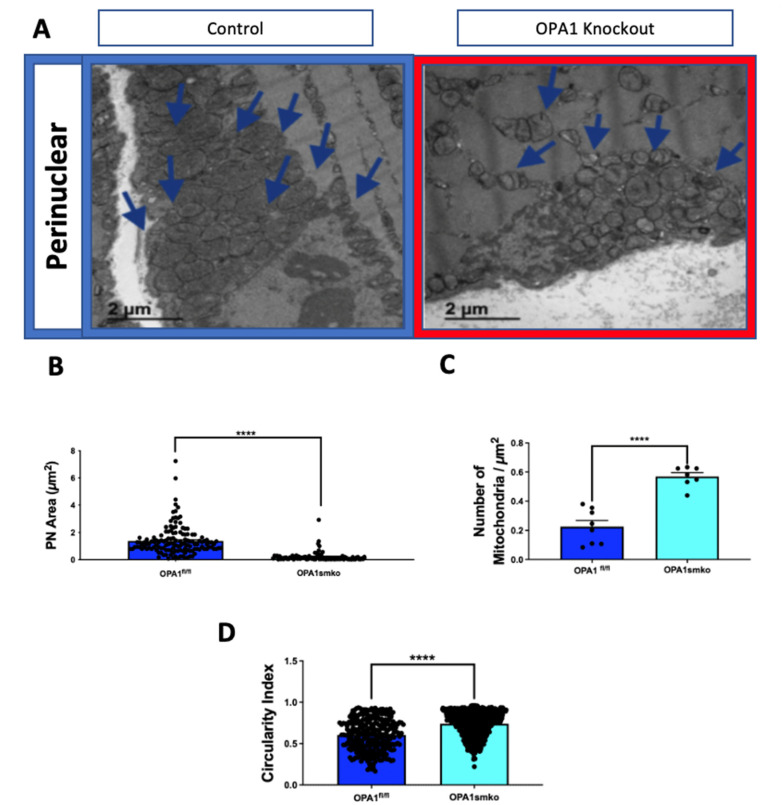
**Knockdown of optic atrophy protein-1** (**OPA1) decreases mitochondrial area.** (**A**) Representative TEM images of mitochondria in mouse skeletal muscle tissue from OPA1 knockdown (red outline) and control (blue outline) tissue. (**B**) Quantification of perinuclear mitochondrial area, (**C**) number of mitochondria per square micron, and (**D**) mitochondrial circularity index. Statistical significance is indicated by asterisks; ***** indicate *p* ≤ 0.0001.

**Figure 4 cells-10-02177-f004:**
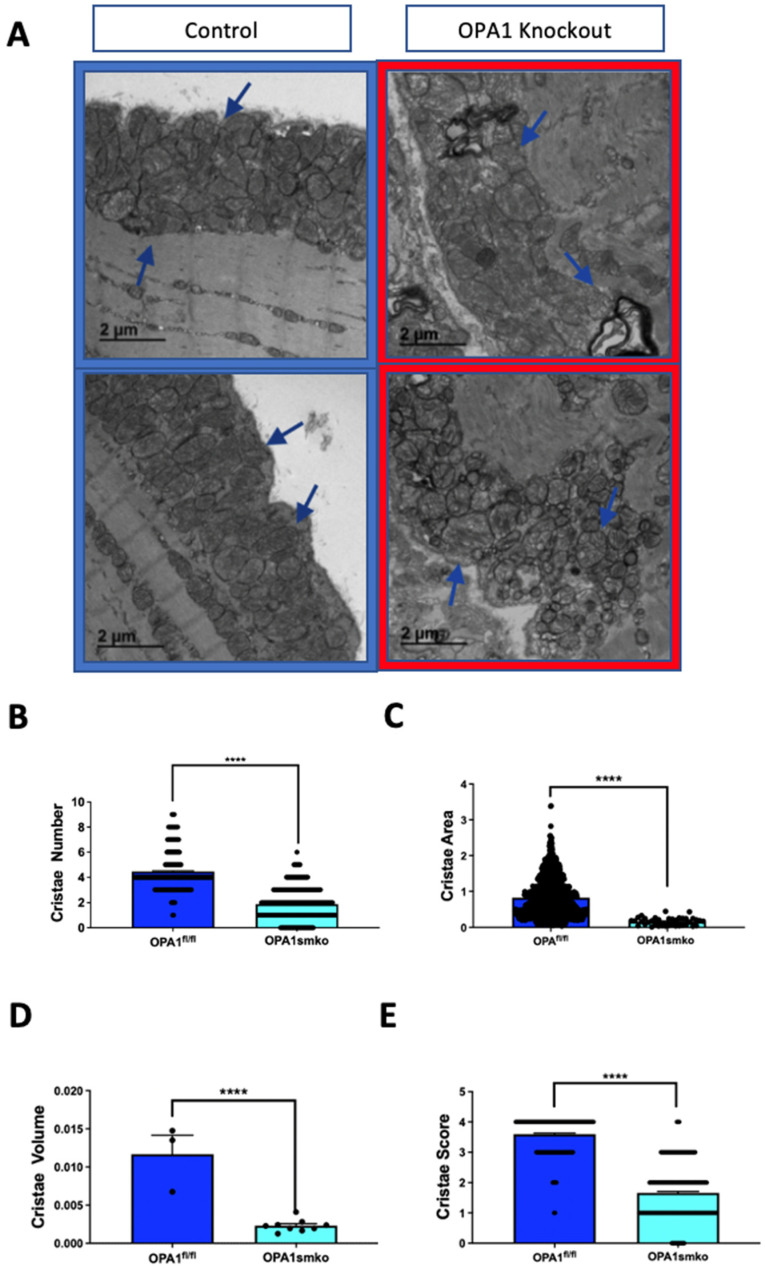
**Knockdown of OPA1 disrupts cristae structure.** (**A**) Representative TEM images of cristae in mouse skeletal muscle tissue from OPA1 knockdown (red outline) and control (blue outline) tissue. (**B**) Quantification of cristae number, (**C**) cristae area, (**D**) cristae volume, and (**E**) cristae score. Statistical significance is indicated by asterisks; **** indicate *p* ≤ 0.0001.

**Figure 5 cells-10-02177-f005:**
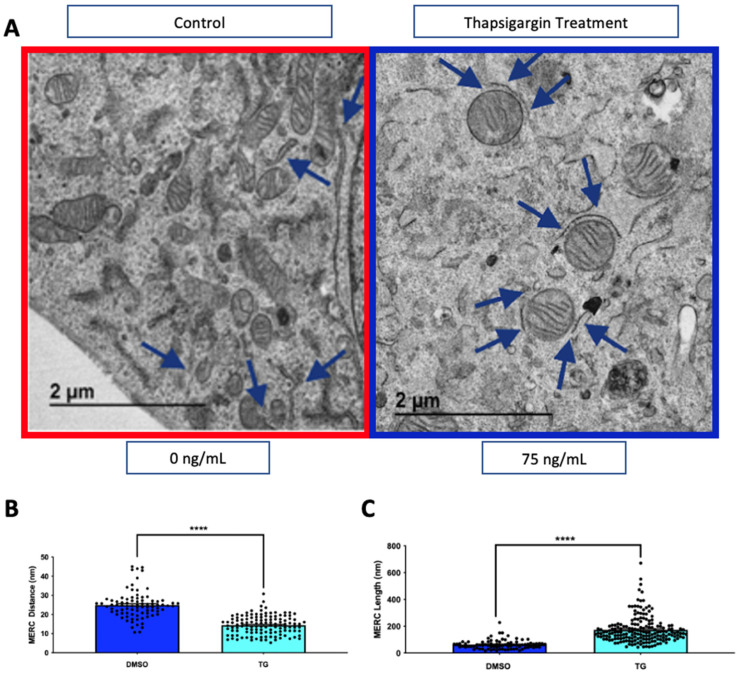
**Thapsigargin treatment increases mitochondria-endoplasmic reticulum contacts (MERCs).** (**A**) Representative TEM images of mouse skeletal myotubes from untreated (DMSO, red outline) and thapsigargin-treated (blue outline) myotubes. MERCs are identified with blue arrows. (**B**) Quantification of MERC distance and (**C**) MERC length. Statistical significance is indicated by asterisks, **** indicate *p* ≤ 0.0001.

## Data Availability

Not applicable.

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
