# Peer review of "A Universal Approach to Analyzing Transmission Electron Microscopy with ImageJ"

_cells, 2021, doi:10.3390/cells10092177_

Round 1
Reviewer 1 Report
The manuscript describes a protocol for measuring features in TEM images using ImageJ software. The protocol is likely to be useful for those who need this type of methods. However, the authors seem to be unaware of previous, well-established methods for quantification of TEM images; stereology and point counting. The authors also completely fail to emphasize the importance of proper sampling during every step of sample preparation. If the TEM images do not accurately represent the samples from which they originate, analyzing the images is pointless since the results are biased no matter how the measurements are done.
The authors must mention the importance of sampling: sampling should be uniform random at every step. For reference on uniform random sampling, please see: Howard, C. V., and Reed, M. G. (2005). ‘‘Unbiased stereology. Three-dimensional measurement in microscopy.’’ Garland Science/BIOS Scientific Publishers, New York.
- Length and width of an organelle like a mitochondrion depend on the orientation of the thin sections in relation to the organelle. Further, the orientation of the organelle may depend on the tissue – e.g. in muscle, mitochondria line along the muscle fibers, and in cultured cells, mitochondria tend to lie horizontally. These aspects should be discussed when describing a methods for measuring organelle length and width.
- Point 1.6.1 is unclear - what is meant by image length?
- Point 3.4. You cannot get ‘area’ from a 2D image - use another word here.
- Point 3.6. You cannot get ‘volume’ from a 2D image using this protocol. Use a more appropriate word here.
- Point 3.7. This classification seems far from being accurate. Which criteria is it based on (numerical or subjective)? Please clarify.
- Point 3.8.1. Unclear: ‘1000 mitochondria from 100 cells’ (total of 1000 or total of 1000 x 100)? ‘10 mitochondria per cell from 2,000 cells’ (total of 20000?). Please clearly state the total number of mitochondria you propose to be analyzed.
- Chapter 5 is unnecessary repetition from the previous sections. Please remove.
- Chapter 6. It remains unclear what is the relationship of pseudo-coloration and the ImageJ analysis. It seems the pseudocoloration is a step that is not necessary for the analysis, but this is not clearly stated.
- Figure 3. The panels from Adobe Photoshop are far too small to be useful. Please increase the size (and perhaps move the Supplement).
- Point 6.8.4. is unclear.
- Figure 6. No autophagosomes can be seen in the TEM panels. I assume the black arrows (which are not explained in the legend) are supposed to indicate autophagic structures, but they are definitely not autophagosomes – some of them may be autolysosomes. The authors must not present misleading images – there are plenty of examples in literature on incorrect identification of autophagic structures. This paper should not be an addition to that list. I suggest removing the ‘autophagosome’ quantification from the manuscript.
- The statement “there is no standardized protocol for quantitatively assessing organelle morphology with TEM.” in Discussion must be removed. The well-established and documented point counting method (with uniform random sampling) has been in use for decades and is still valid. In addition, in many cases point counting is faster than tracing the features by hand, as instructed in the present manuscript. For reference on point counting, see e.g. Howard, C. V., and Reed, M. G. (2005). ‘‘Unbiased stereology. Three-dimensional measurement in microscopy.’’ Garland Science/BIOS Scientific Publishers, New York. Of note, the present protocol brings most advantage for analysis of features like mitochondrial cristae.
- Line 345: “To address this, higher magnifications of 2,500x or even 5,000x should be used to better resolve organelles of interest” The final magnification of the TEM image depends on the location of camera in the TEM – bottom-mounted camera gives a much higher final magnification than a side-mounted camera, when the same magnification is used to take the image. Therefore it is misleading to mention accurate numbers for the magnification that should be used for taking the images.
- Line 354: “organelle volume or organelle interactions in three dimensions cannot be reliably assessed without using alternative methods. In such cases, more advanced techniques, including focused ion beam scanning electron microscopy (FIB-SEM) or serial block-face scanning electron microscopy (SBF-SEM), can be used to image subcellular structures in three dimensions” This is inaccurate and misleading. Point counting can be used to measure organelle volume and organelle number using conventional TEM images. For reference, see e.g. Howard, C. V., and Reed, M. G. (2005). ‘‘Unbiased stereology. Three-dimensional measurement in microscopy.’’ Garland Science/BIOS Scientific Publishers, New York.
Author Response
Dear Editor,
Thank you for allowing us to submit a revised version of our manuscript entitled “A Universal Approach to Analyzing Transmission Electron Microscopy with ImageJ” for publication in Cells. We appreciate the time and effort that you and the reviewers dedicated to providing feedback on our manuscript. The insightful comments have tremendously improved our manuscript. The suggestions made by the reviewers necessitated a revision of the figures, methods, and discussion sections.
Below you will find our responses to the reviewers’ comments. Our responses have been intentionally made in blue. All page numbers refer to the revised manuscript file with tracked changes. We hope that in the present form, the revised manuscript will be acceptable for publication.
Reviewer 1:
- Length and width of an organelle like a mitochondrion depend on the orientation of the thin sections in relation to the organelle. Further, the orientation of the organelle may depend on the tissue – e.g. in muscle, mitochondria line along the muscle fibers, and in cultured cells, mitochondria tend to lie horizontally. These aspects should be discussed when describing a methods for measuring organelle length and width.
We thank the reviewer for noting that our methods, as written, requires more description. We have therefore revised the manuscript to point out that Length and width is the same for all tissues. Based upon looking at a muscle myotubes in cell culture, these things may be similar and lay horizontally. However, in the context of other types of cell lines, the mitochondria may be in another direction. It is important to consider point to point, where the cell is being measured. In the methods section, we added the definition of the calculations and added equations to find that particular measurement. We hope this has adequately clarified our quantifications for the future reproduction of this protocol (lines 410-450).
- Point 1.6.1 is unclear - what is meant by image length?
We again thank the reviewer for noting that clarity is needed for image length. Image length can be recorded according to Feret’s diameter. A clarification that this was referring to bar length was added (lines 101-102). Definition and calculation of how this can be done has been added to the Methods to address this (lines 419-422).
- Point 3.4. You cannot get ‘area’ from a 2D image - use another word here.
We thank the reviewer for this comment. To address this point, definition and calculation of how this can be done has been added to the Methods (lines 423-428). Image J can be used to measure area by using pixel intensity and can be found in the following citations, which we have also cited:
Schneider CA, Rasband WS, Eliceiri KW. NIH Image to ImageJ: 25 years of image analysis. Nat Methods. 2012 Jul;9(7):671-5. doi: 10.1038/nmeth.2089. PMID: 22930834; PMCID: PMC5554542.
Schindelin J, Arganda-Carreras I, Frise E, Kaynig V, Longair M, Pietzsch T, Preibisch S, Rueden C, Saalfeld S, Schmid B, Tinevez JY, White DJ, Hartenstein V, Eliceiri K, Tomancak P, Cardona A. Fiji: an open-source platform for biological-image analysis. Nat Methods. 2012 Jun 28;9(7):676-82. doi: 10.1038/nmeth.2019. PMID: 22743772; PMCID: PMC3855844.
This is a true measurement and has been shown and done before in previous papers. Please see following papers:
Ponce JM, Coen G, Spitler KM, Dragisic N, Martins I, Hinton A Jr, Mungai M, Tadinada SM, Zhang H, Oudit GY, Song LS, Li N, Sicinski P, Strack S, Abel ED, Mitchell C, Hall DD, Grueter CE. Stress-Induced Cyclin C Translocation Regulates Cardiac Mitochondrial Dynamics. J Am Heart Assoc. 2020 Apr 7;9(7):e014366. doi: 10.1161/JAHA.119.014366. Epub 2020 Apr 4. PMID: 32248761; PMCID: PMC7428645.
Carter CS, Huang SC, Searby CC, Cassaidy B, Miller MJ, Grzesik WJ, Piorczynski TB, Pak TK, Walsh SA, Acevedo M, Zhang Q, Mapuskar KA, Milne GL, Hinton AO Jr, Guo DF, Weiss R, Bradberry K, Taylor EB, Rauckhorst AJ, Dick DW, Akurathi V, Falls-Hubert KC, Wagner BA, Carter WA, Wang K, Norris AW, Rahmouni K, Buettner GR, Hansen JM, Spitz DR, Abel ED, Sheffield VC. Exposure to Static Magnetic and Electric Fields Treats Type 2 Diabetes. Cell Metab. 2020 Oct 6;32(4):561-574.e7. doi: 10.1016/j.cmet.2020.09.012. Erratum in: Cell Metab. 2020 Dec 1;32(6):1076. PMID: 33027675; PMCID: PMC7819711.
Parra V, Verdejo HE, Iglewski M, Del Campo A, Troncoso R, Jones D, Zhu Y, Kuzmicic J, Pennanen C, Lopez-Crisosto C, Jaña F, Ferreira J, Noguera E, Chiong M, Bernlohr DA, Klip A, Hill JA, Rothermel BA, Abel ED, Zorzano A, Lavandero S. Insulin stimulates mitochondrial fusion and function in cardiomyocytes via the Akt-mTOR-NFκB-Opa-1 signaling pathway. Diabetes. 2014 Jan;63(1):75-88. doi: 10.2337/db13-0340. Epub 2013 Sep 5. PMID: 24009260; PMCID: PMC3868041.
Tsushima K, Bugger H, Wende AR, Soto J, Jenson GA, Tor AR, McGlauflin R, Kenny HC, Zhang Y, Souvenir R, Hu XX, Sloan CL, Pereira RO, Lira VA, Spitzer KW, Sharp TL, Shoghi KI, Sparagna GC, Rog-Zielinska EA, Kohl P, Khalimonchuk O, Schaffer JE, Abel ED. Mitochondrial Reactive Oxygen Species in Lipotoxic Hearts Induce Post-Translational Modifications of AKAP121, DRP1, and OPA1 That Promote Mitochondrial Fission. Circ Res. 2018 Jan 5;122(1):58-73. doi: 10.1161/CIRCRESAHA.117.311307. Epub 2017 Nov 1. PMID: 29092894; PMCID: PMC5756120.
- Point 3.6. You cannot get ‘volume’ from a 2D image using this protocol. Use a more appropriate word here.
We thank the reviewer for this comment. More recently, it has been shown in previous papers that ‘volume’ is acceptable, but it is not “volume” in the traditional sense (lines 437-438). 2D volume is akin to an estimation and is not the most accurate measurement, but others have recorded it. This calculation can be done, but we do not recommend using this. To count something in a volumetric way in 2D, the point measure is the best option (like 337-346). Stereology must be performed to find the volume. We have made it clearer that this is an estimate. Here are the following citations that have measured this in the past:
Pamenter ME, Perkins GA, Gu XQ, Ellisman MH, Haddad GG. DIDS (4,4-diisothiocyanatostilbenedisulphonic acid) induces apoptotic cell death in a hippocampal neuronal cell line and is not neuroprotective against ischemic stress. PLoS One. 2013;8(4):e60804. doi: 10.1371/journal.pone.0060804. Epub 2013 Apr 5. PMID: 23577164; PMCID: PMC3618322.
Shults NV, Kanovka SS, Ten Eyck JE, Rybka V, Suzuki YJ. Ultrastructural Changes of the Right Ventricular Myocytes in Pulmonary Arterial Hypertension. J Am Heart Assoc. 2019 Mar 5;8(5):e011227. doi: 10.1161/JAHA.118.011227. PMID: 30807241; PMCID: PMC6474942.
Amini P, Stojkov D, Felser A, Jackson CB, Courage C, Schaller A, Gelman L, Soriano ME, Nuoffer JM, Scorrano L, Benarafa C, Yousefi S, Simon HU. Neutrophil extracellular trap formation requires OPA1-dependent glycolytic ATP production. Nat Commun. 2018 Jul 27;9(1):2958. doi: 10.1038/s41467-018-05387-y. PMID: 30054480; PMCID: PMC6063938.
Patra M, Mahata SK, Padhan DK, Sen M. CCN6 regulates mitochondrial function. J Cell Sci. 2016 Jul 15;129(14):2841-51. doi: 10.1242/jcs.186247. Epub 2016 Jun 1. PMID: 27252383.
Ware C. I. (2003). A simple method to 'point count' silt using scanning electron microscopy aided by image analysis. Journal of microscopy, 212(Pt 2), 205–208. https://doi.org/10.1046/j.1365-2818.2003.01234.x
- Point 3.7. This classification seems far from being accurate. Which criteria is it based on (numerical or subjective)? Please clarify.
We again, thank the reviewer for this comment, and we will gladly clarify this point. Eisner et al., developed a cristate score classification that has been previously published & referenced in the manuscript. The reference for the paper by Eisner et al., is below:
Eisner V, Cupo RR, Gao E, Csordás G, Slovinsky WS, Paillard M, Cheng L, Ibetti J, Chen SR, Chuprun JK, Hoek JB, Koch WJ, Hajnóczky G. Mitochondrial fusion dynamics is robust in the heart and depends on calcium oscillations and contractile activity. Proc Natl Acad Sci U S A. 2017 Jan 31;114(5):E859-E868. doi: 10.1073/pnas.1617288114. Epub 2017 Jan 17. PMID: 28096338; PMCID: PMC5293028.
- Point 3.8.1. Unclear: ‘1000 mitochondria from 100 cells’ (total of 1000 or total of 1000 x 100)? ‘10 mitochondria per cell from 2,000 cells’ (total of 20000?). Please clearly state the total number of mitochondria you propose to be analyzed.
We thank the reviewer for noting that clarification is needed for the number of mitochondria. These amounts are based on results from Parra et al (referenced below). This has been clarified (lines 144-146) to show that either of these two amounts may be utilized to obtain a representative sample. Additionally, both of these methods average the same number of mitochondria, although different collection protocols were utilized (either on the basis of an average per cell or a total from a group of cells).
Parra, V., Verdejo, H. E., Iglewski, M., Del Campo, A., Troncoso, R., Jones, D., Zhu, Y., Kuzmicic, J., Pennanen, C., Lopez-Crisosto, C., Jaña, F., Ferreira, J., Noguera, E., Chiong, M., Bernlohr, D. A., Klip, A., Hill, J. A., Rothermel, B. A., Abel, E. D., Zorzano, A., … Lavandero, S. (2014). Insulin stimulates mitochondrial fusion and function in cardiomyocytes via the Akt-mTOR-NFκB-Opa-1 signaling pathway. Diabetes, 63(1), 75–88. https://doi.org/10.2337/db13-0340
- Chapter 5 is unnecessary repetition from the previous sections. Please remove.
We thank the reviewer for pointing out that chapter 5 should be removed. We have considered the reviewer’s helpful comment and have removed the section.
- Chapter 6. It remains unclear what is the relationship of pseudo-coloration and the ImageJ analysis. It seems the pseudocoloration is a step that is not necessary for the analysis, but this is not clearly stated.
We thank the reviewer for pointing out that our pseudocoloring section, as written, is a bit unclear. Pseudocoloring is not needed to measure ImageJ analysis. It is mainly needed to look at spatial relationship and to designate individual colors or organelles. This can be done in both ImageJ and Photoshop. However, photoshop is quicker and more automated. This is addressed in the conclusion (lines 326-329).
- Figure 3. The panels from Adobe Photoshop are far too small to be useful. Please increase the size (and perhaps move the Supplement).
We thank the reviewer for pointing out that our panels are too small and illegible. This has been fixed. We have ensured the descriptions for each panel is clear in the step-wish process. We have changed the figure to include ImageJ instead (as a supplement) and we have changed the Adobe Photoshop figure to the supplement as well, as recommended.
- Point 6.8.4. is unclear.
We thank the reviewer for the comment and suggestion. This has been addressed. We have taken it out completely and is no longer in the text as it was not strictly necessary and beyond the scope of this manuscript.
- Figure 6. No autophagosomes can be seen in the TEM panels. I assume the black arrows (which are not explained in the legend) are supposed to indicate autophagic structures, but they are definitely not autophagosomes – some of them may be autolysosomes. The authors must not present misleading images – there are plenty of examples in literature on incorrect identification of autophagic structures. This paper should not be an addition to that list. I suggest removing the ‘autophagosome’ quantification from the manuscript.
We thank the reviewer for the comment and suggestions. Given the nuances of autophagosome and lysosome identification (references below), we have decided that the explanation of proper identification is beyond the scope of this paper. Therefore, we have removed these figures and will submit this data in a separate manuscript.
Eskelinen, E. L. Maturation of autophagic vacuoles in mammalian cells. Autophagy 5, 1-10 (2005).
Eskelinen, E. L. To be or not to be? Examples of incorrect identification of autophagic compartments in conventional transmission electron microscopy of mammalian cells. Autophagy 4, 257-260 (2008).
Eskelinen, E. L., Reggiori, F., Baba, M., Kovács, A. L., & Seglen, P. O. Seeing is believing: the impact of electron microscopy on autophagy research. Autophagy 7, 935-956 (2011).
- The statement “there is no standardized protocol for quantitatively assessing organelle morphology with TEM.” in Discussion must be removed. The well-established and documented point counting method (with uniform random sampling) has been in use for decades and is still valid. In addition, in many cases point counting is faster than tracing the features by hand, as instructed in the present manuscript. For reference on point counting, see e.g. Howard, C. V., and Reed, M. G. (2005). ‘‘Unbiased stereology. Three-dimensional measurement in microscopy.’’ Garland Science/BIOS Scientific Publishers, New York. Of note, the present protocol brings most advantage for analysis of features like mitochondrial cristae.
We thank the reviewer for this comment and the point is well taken. Point taking is a useful method that has many applications, especially for analysis of complete images. There are, however, limitations to both methods. Point counting is an estimation. Hand tracing, although more labor intensive, is comparable to point counting. The biggest advantage is the better imaging of cristae, which we have made clearer. This comes at a cost of requiring more time to perform ImageJ analysis. We suggest both must be done to assess organelle morphology with TEM. We postulate that 3D is the best, but we understand that not everyone has access to 3D imaging; therefore, the discussion has been changed to better address these considerations (lines 306-312; lines 335-350).
- Line 345: “To address this, higher magnifications of 2,500x or even 5,000x should be used to better resolve organelles of interest” The final magnification of the TEM image depends on the location of camera in the TEM – bottom-mounted camera gives a much higher final magnification than a side-mounted camera, when the same magnification is used to take the image. Therefore it is misleading to mention accurate numbers for the magnification that should be used for taking the images.
We thank the reviewer for this suggestion. We have strongly considered and have revised the text to better clarify and not state specific magnifications (line 325-326).
- Line 354: “organelle volume or organelle interactions in three dimensions cannot be reliably assessed without using alternative methods. In such cases, more advanced techniques, including focused ion beam scanning electron microscopy (FIB-SEM) or serial block-face scanning electron microscopy (SBF-SEM), can be used to image subcellular structures in three dimensions” This is inaccurate and misleading. Point counting can be used to measure organelle volume and organelle number using conventional TEM images. For reference, see e.g. Howard, C. V., and Reed, M. G. (2005). ‘‘Unbiased stereology. Three-dimensional measurement in microscopy.’’ Garland Science/BIOS Scientific Publishers, New York.
We again thank the reviewer for this comment and suggestions. Point counting is a useful method that we neglected to adequately mention in the prior manuscript. We do understand that point counting is a widely accepted method, and in the revision we have better discussed this. In addition, other methodologies, such as hand tracing is another accurate method for counting and measuring organelles, especially when it comes to features such as cristae. We have expanded the discussion of both (lines 306-312; lines 335-350).
Reviewer 2 Report
This is a methodological paper describing the use of ImageJ analysis on TEM images for the evaluation of mitochondria morphology and parameters.
I think that this paper is of value for the scientific community.
However some point need to be addressed before publication.
1) please describe how the specimen used for the analysis was obtained. Especially for the murine skeletal muscle cells that were not further described. This is important to allow complete replication of the procedure.
2) use of commercial software like adobe photo shop is not appropriate, especially considering that ImageJ can perform the same task. Please describe the pseudo colors alteration with ImageJ.
Author Response
Dear Editor,
Thank you for allowing us to submit a revised version of our manuscript entitled “A Universal Approach to Analyzing Transmission Electron Microscopy with ImageJ” for publication in Cells. We appreciate the time and effort that you and the reviewers dedicated to providing feedback on our manuscript. The insightful comments have tremendously improved our manuscript. The suggestions made by the reviewers necessitated a revision of the figures, methods, and discussion sections.
Below you will find our responses to the reviewers’ comments. Our responses have been intentionally made in blue. All page numbers refer to the revised manuscript file with tracked changes. We hope that in the present form, the revised manuscript will be acceptable for publication.
Reviewer 2:
1) please describe how the specimen used for the analysis was obtained. Especially for the murine skeletal muscle cells that were not further described. This is important to allow complete replication of the procedure.
We thank the reviewer for the suggestion, and we agree that a better description is needed. This has been fixed. We have added a description of methods for isolation and care of these specimens (lines 361-408).
2) use of commercial software like adobe photo shop is not appropriate, especially considering that ImageJ can perform the same task. Please describe the pseudo colors alteration with ImageJ.
We again thank the reviewer for the wonderful suggestion. This has been addressed. We have added how this can be done in ImageJ to accompany the alternative Adobe Photoshop protocol to allow for researchers to have as many options as possible (lines 174-207; 326-329).
Round 2
Reviewer 1 Report
The revision has improved the manuscript, but there are still important issued that must be improved. These points (of which No 1 an 2 were already mentioned in my previous report) MUST be addressed:
- The authors completely fail to emphasize the importance of proper sampling during every step of sample preparation. If the TEM images do not accurately represent the samples from which they originate, analyzing the images is pointless since the results are biased no matter how the measurements are done. The authors must mention the importance of sampling: sampling should be uniform random at every step (starting from dissecting the tissue from animal, selecting tissue pieces for EM embedding, selecting epon blocks for sectioning, selecting sections for imaging, selecting areas of sections for taking images). For reference on uniform random sampling, please see: Howard, C. V., and Reed, M. G. (2005). ‘‘Unbiased stereology. Three-dimensional measurement in microscopy.’’ Garland Science/BIOS Scientific Publishers, New York.
- Length and width of an organelle like a mitochondrion depend on the orientation of the thin sections in relation to the organelle (in other words, are e.g. the mitochondria visible in the thin sections as cross sections or longitudinal sections). Further, the orientation of the organelle may depend on the tissue – e.g. in muscle, mitochondria line along the muscle fibers, and in cultured cells, mitochondria tend to lie horizontally. Thus, mitochondrial size and shape ofter depends on how the tissue or cells are sectioned (cross sections or longitudinal sections). These aspects should be discussed when describing a methods for measuring organelle length and width. Importantly, the response the authors provided in the rebuttal does not address the issue I raised here.
- Line 75: “There is no clear consensus regarding the most effective methods for measuring organelle features from TEM images.” Remove this sentence, it is not really true.
- Line 306-3-307: “However, to our knowledge, there is no standardized protocol for quantitatively assessing finer structure organelle morphology, such as cristae, with TEM.” Remove this sentence, it is not really true.
- Line 339: “Alternatively, in the past, point counting has been used and has been demonstrated to be very useful and accurate.” Remove ‘in the past’ as point counting is still used by numerous researchers using TEM.
Author Response
Manuscript cells-1284131
Response to Reviewers
Dear Editor,
Thank you for allowing us to submit a revised version of our manuscript entitled “A Universal Approach to Analyzing Transmission Electron Microscopy with ImageJ” for publication in Cells. We appreciate the time and effort that you and the reviewers dedicated to providing feedback on our manuscript. The insightful comments have tremendously improved our manuscript. The suggestions made by the reviewers necessitated a revision of the discussion and perspective sections.
Below you will find our responses to the reviewers’ comments. Our responses have been intentionally made in blue. All page numbers refer to the revised manuscript file with tracked changes. We hope that in the present form, the revised manuscript will be acceptable for publication.
Reviewer 1:
- Line 75: “There is no clear consensus regarding the most effective methods for measuring organelle features from TEM images.” Remove this sentence, it is not really true.
We thank the reviewer for this suggestion and have removed this sentence as advised. (lines 73-74).
- Line 306-307: “However, to our knowledge, there is no standardized protocol for quantitatively assessing finer structure organelle morphology, such as cristae, with TEM.” Remove this sentence, it is not really true.
We thank the reviewer for this comment and have removed this sentence as suggested (lines 306-307).
- Line 339: “Alternatively, in the past, point counting has been used and has been demonstrated to be very useful and accurate.” Remove ‘in the past’ as point counting is still used by numerous researchers using TEM.
We thank the reviewer for this suggestion and have altered this sentence to reflect better that point counting has both been used effectively in the past and will continue to be used effectively in the future (lines 341-342).
- The authors completely fail to emphasize the importance of proper sampling during every step of sample preparation. If the TEM images do not accurately represent the samples from which they originate, analyzing the images is pointless since the results are biased no matter how the measurements are done. The authors must mention the importance of sampling: sampling should be uniform random at every step (starting from dissecting the tissue from animal, selecting tissue pieces for EM embedding, selecting epon blocks for sectioning, selecting sections for imaging, selecting areas of sections for taking images). For reference on uniform random sampling, please see: Howard, C. V., and Reed, M. G. (2005). ‘‘Unbiased stereology. Three-dimensional measurement in microscopy.’’ Garland Science/BIOS Scientific Publishers, New York.
We thank the reviewer for raising this important concern. Proper sampling is of utmost importance, and we have modified our manuscript to better take this into account. In addition to the previously added methods section regarding proper randomness during image analysis, (lines 468-474) we have additionally added a section titled “PERSPECTIVES ON TEM STANDARDIZATION.” (lines 379-408). In this section, we discuss how our protocol has ensured uniform random sampling throughout while giving recommendations on how to avoid bias in every step of the experimental process from tissue sampling to analyzing the data. This section takes into account the points raised by Howard, C. V., and Reed, M. G. (2004) along with points raised in:
Mouton, P. R. (2011). Applications of unbiased stereology to neurodevelopmental toxicology. Developmental Neurotoxicology Research. Hoboken, NJ: Wiley, 53-75.
Wernitznig, S., Reichmann, F., Sele, M., Birkl, C., Haybäck, J., Kleinegger, F., ... & Leitinger, G. An Unbiased Approach of Sampling TEM Sections in Neuroscience. JoVE (Journal of Visualized Experiments) 2019, 146, e58745.
We thank you again for bringing our attention to this omission and we hope we have adequately elucidated this topic for future researchers.
- Length and width of an organelle like a mitochondrion depend on the orientation of the thin sections in relation to the organelle (in other words, are e.g. the mitochondria visible in the thin sections as cross sections or longitudinal sections). Further, the orientation of the organelle may depend on the tissue – e.g. in muscle, mitochondria line along the muscle fibers, and in cultured cells, mitochondria tend to lie horizontally. Thus, mitochondrial size and shape ofter depends on how the tissue or cells are sectioned (cross sections or longitudinal sections). These aspects should be discussed when describing a methods for measuring organelle length and width. Importantly, the response the authors provided in the rebuttal does not address the issue I raised here.
We thank the reviewer for raising this important concern. We apologize for not adequately addressing it in the previous rebuttal and we have made more extensive edits to address this in the section “PERSPECTIVES ON TEM MEASUREMENTS”. We bring up the points raised, along with alternative measurements which may be performed if one is unsure of the length and orientation of a mitochondrion (lines 365-378).